

# Differentiation patterns of emperor moths (Lepidoptera: Saturniidae: Saturniinae) of a continental island: divergent evolutionary history driven by Pleistocene glaciations

Wen-Bin Yeh[1], Cheng-Lung Tsai[1], Thai-Hong Pham[2,3], Shipher Wu[4], Chia-Wei Chang[1] and Hong-Minh Bui[5]

[1] Department of Entomology, National Chung Hsing University, Taichung, Taiwan
[2] Vietnam National Museum of Nature and Graduate University of Science and Technology, Vietnam Academy of Science and Technology, Hanoi, Vietnam
[3] Mientrung Institute for Scientific Research, Vietnam Academy of Science and Technology, Hue, Vietnam
[4] Biodiversity Research Center, Academia Sinica, Taipei, Taiwan
[5] Hanoi National University of Education, Hanoi, Vietnam

Corresponding author
Wen-Bin Yeh, wbyeh@nchu.edu.tw

## ABSTRACT

**Background:** On the basis of molecular dating, Pleistocene glaciations have been proposed as the major driving force of biota speciation in the Palearctic and the pre-Quaternary origin of Amazonian taxa. However, the major driving factors in East Asia remain unclear. All 16 saturniine species inhabiting Taiwan with congeners of populations, subspecies, or species in East Asia constitute research objects for addressing the mode of speciation because of the repeated formation and disappearance of a landbridge from the Asian mainland to Taiwan during glacial cycles.

**Methods:** The genetic divergences of mitochondrial cytochrome c oxidase subunit I (COI) and 16S rDNA and the nuclear 28S rDNA of the saturniine species from Taiwan and the Asian mainland were assessed to determine the monophyly of each genus and species of Saturniinae. Moreover, 519 saturniine COI sequences of 114 taxa from adjacent East and Southeast Asian populations and closely related species were retrieved from GenBank and analyzed. The differentiation timing and possible origination of the insular saturniines were elucidated based on phylogenetic relationships, haplotype networks, and lineage calibrations.

**Results:** Approximately 90% of intraspecific COI divergence was <2%; all divergences exceeding 2% originated from comparisons between allopatric populations or subspecies. Relationship analyses revealed that multiple introductions likely occurred in insular saturniines and that some East Asian saturniines were paraphyletic as deduced by analyzing endemic insular species. Calibration dating revealed that Taiwanese endemic saturniines split from sibling Asian species 0.2–2.7 million years ago (Mya), whereas subspecific-level and population-level splitting events occurred 0.1–1.7 Mya and 0.2–1.2 Mya, respectively. Moreover, phylogenetic patterns combined with geographical distributions revealed that hill-distributed Taiwanese saturniines are closely related to those from southern
China and Southeast Asia, whereas saturniines inhabiting altitudes higher than 1,500 m in Taiwan have siblings distributed in temperate Northeast Asia.

**Discussion:** The Global DNA Barcoding Initiative was successfully applied to study the population genetic structure in species. Most Formosan saturniines are distinct and monophyletic, reflecting the vicariant barrier of the Taiwan Strait; Pleistocene glacial cycles provided opportunities for insular saturniines to experience repeated isolation from and secondary contact with the continental mainland. Each insular saturniine may have evolved with a unique differentiation timing pattern that possibly emerged in the Early, Middle, or Late Pleistocene with these patterns differing from the consistent pattern that occurred in the temperate Palearctic and tropical Amazonian regions. Moreover, multiple migrations or artificial genetic admixtures may have also occurred, as suggested by the coexistence of two divergent lineages in a few Taiwanese saturniines.

# INTRODUCTION

Pleistocene climatic fluctuation had profound effects on the origins and diversification of extant organisms (*Avise, 2000*; *Hewitt, 2000*; *Hewitt, 2004*). During glaciation, decreasing temperatures forced organisms living in high-latitude or high-altitude locations to migrate southward to warmer areas or to lower elevations, respectively, and then migrate back during interglacial periods (*Hewitt, 2000*; *Tribsch & Schönswetter, 2003*; *Hewitt, 2004*; *Lohse, Nicholls & Stone, 2011*). Molecular dating has revealed that the speciation of organisms in continental Europe and North America can be dated back to 1–2 million years ago (Mya), which was a time of Pleistocene glacial cycles (*Avise, 2000*; *Hewitt, 2000*; *Knowles, 2000*; *Willis & Whittaker, 2000*; *Hewitt, 2004*). In Amazonian and Andean regions, however, the major time period for the origins of species may be the pre-Quaternary period, defined as >2.6 Mya (*Willis & Whittaker, 2000*; *Rull, 2008*; *Hoorn et al., 2010*; *Rull, 2011*). Moreover, for distinct phylogeographical subspecies and intraspecific lineages, the divergence times proposed for European fishes and Holarctic organisms are primarily within 0.1–1 Mya (*Hewitt, 2004*).

Pleistocene glacial cycles also had a major influence on organism speciation in East Asia, especially for continental islands such as Taiwan because of the repeated formation and disappearance of a land bridge from the Asian mainland. Four hypotheses concerning population differentiation on the island of Taiwan have been proposed (*Tsai, Wan & Yeh, 2014*), but the factors driving the divergent speciation of Taiwan-inhabiting organisms compared with their mainland counterparts remain unclear. The Pleistocene hypothesis has been shown to be valid for *Cylindera* tiger beetles, *Euphaea* damselflies, *Nanhaipotamon* crabs, stage beetles, and *Takydromus* lizards (*Lin, Chen & Lue, 2002*; *Shih et al., 2011*; *Sota et al., 2011*; *Lee & Lin, 2012*; *Tsai, Wan & Yeh, 2014*; *Tsai & Yeh, 2016*). However, molecular dating of *Hynobius* salamanders, *Pyrgonota*

treehoppers, and *Potamid* crabs has indicated that origins of >2.6 Mya accord with the pre-Quaternary hypothesis (*Yeo et al., 2007*; *Shih, Yeo & Ng, 2009*; *Li, Fu & Lei, 2011*; *Su et al., 2014*; *Su et al., 2016*).

Taiwan, previously known as Formosa, is a mountainous subtropical island situated near the coast of the southeastern Asian mainland and north of the Philippines. It was formed from the collision of the Eurasian plate and Philippine Sea Plate at approximately 6 Mya (*Su, 1984*; *Huang et al., 1997*; *Sibuet & Hsu, 2004*). A drastic increase in mountain height at approximately 1–2.5 Mya (*Teng, 1990*; *Huang et al., 1997*) resulted in the formation of the Central Mountain Range, with the highest peak altitude being 3,952 m. During Pleistocene glacial events, the decline of sea levels resulted in land bridge formation between Taiwan and mainland Asia, which provided opportunities for biota to migrate across the Taiwan Strait (*Lue & Chen, 1997*; *Voris, 2000*; *Lin, Chen & Lue, 2002*; *Huang, 2006*; *Jang-Liaw, Lee & Chou, 2008*; *Wu et al., 2010*; *Huang & Lin, 2011*; *Tsai, Wan & Yeh, 2014*; *Tsai & Yeh, 2016*; *Weng, Yang & Yeh, 2016*; *Kurita et al., 2017*; *Niu et al., 2018*).

Molecular dating has demonstrated that periodic glaciation events introduced various organisms to Taiwan. The cricket *Loxoblemmus appendicularis* Shiraki, stag beetles, and alpine carabids arrived in Taiwan 1.0–2.8 Mya and have been classified as several deeply divergent geographical lineages of species and subspecies (*Yeh et al., 2004*; *Tsai, Wan & Yeh, 2014*; *Tsai & Yeh, 2016*; *Weng, Yang & Yeh, 2016*; *Weng, Yeh & Yang, 2016*). Conversely, the birdwing butterfly *Troides aeacus* (C. & R. Felder), the ground beetle *Nebria uenoiana* Habu, and several cyprinid fishes exhibiting shallow or no divergence are species from Asia that migrated during the most recent glacial maximum (*Lin et al., 2010*; *Wu et al., 2010*; *Yang et al., 2012*; *Weng, Yang & Yeh, 2016*). Considerably more complex introduction patterns have been discovered for toads, tiger beetles, fir pines, and tiger shrimp, which migrated to Taiwan from multiple origins during the Pleistocene glacial cycles (*Chung et al., 2004*; *You et al., 2008*; *Sota et al., 2011*; *Yu, Lin & Weng, 2014*).

The colorful Saturniidae (Lepidoptera) family comprises approximately 3,500 species (*Kitching et al., 2018*), including many economically critical species and silk-producing species such as the atlas moth, emperor moth, and luna moth. All 16 saturniine species of Taiwan with congeners of populations, subspecies, or species in East and Southeast Asia constitute ideal research objects for addressing how speciation occurs on subtropical Taiwan. The morphological variations of the isolated populations on Taiwan and adjacent mainland and islands expose the divergent differentiation history of several saturniine species. For example, researchers have long suggested that *Actias sinensis subaurea* Kishida —with its small size and well-developed, broad dark submarginal band on its forewing— should be regarded as its own species distinct from *Actias sinensis* (Walker) (*Kishida, 1993*; *d'Abrera, 1998*). The spectacular emperor moth *Attacus atlas formosanus* Villiard has long been recognized as a distinct subspecies of *Attacus atlas* (Lin.) from Taiwan (*Peigler & Wang, 1996*), whereas *Kitching et al. (2018)* suggested that there is no subspecific recognition among *Att. atlas* populations (*Pinratana & Lampe, 1990*; *Peigler & Wang, 1996*; *Zhu & Wang, 1996*).

Repeated dispersals during the Pleistocene glacial cycles provided spatiotemporal opportunities for repeated allopatric isolation and secondary contact. Deep, middle, or

shallow genetic divergences that possibly occurred in the Early, Middle, or Late Pleistocene could be expected between closely related lineages of saturniine in Taiwan and mainland Asia. The single introduction hypothesis would suggest that populations in Taiwan are monophyletic; by contrast, if divergent lineages in Taiwan exhibit a pattern of mixing with neighboring lineages, this would agree with the multiple origination scenario.

For geographic origin inferences, some saturniines in Taiwan, such as *Saturnia fukudai* and *Antheraea superba*, inhabit montane areas with closely related species distributed in temperate Japan and Korea, and the hill-distributed *Att. atlas* is widely spread in tropical regions from India and Nepal eastward to southern China and the Philippines. The evolutionary routes of Taiwanese saturniines could therefore be addressed. Moreover, the aforementioned differentiation populations between Taiwan island and Asian mainland indicate that the saturniines despite their large size might be poor flyers, hindering their dispersal across the Taiwan Strait.

In the present study, the genetic divergences of two mitochondrial genes, namely cytochrome c oxidase subunit I (COI) and 16S rDNA, and one nuclear gene, namely 28S rDNA, of the saturniine species from Taiwan were assessed to determine the monophyly of each genus and species of Saturniinae. Moreover, the global DNA barcoding project provides many saturniine COI sequences, which are also commonly used in calibration dating, from adjacent Asian populations and closely related species, thus enabling this study to comprehensively address the possible origins and divergence times of each insular saturniine. On the basis of the COI sequence divergences, phylogenetic monophyly, haplotype networks, and lineage calibrations of these saturniines, the origin of each insular saturniine from their closely related mainland lineages during different periods or glaciation events was elucidated. We aimed to determine the speciation mode of the continental subtropical island and to compare whether the mode can be reconciled with the Pleistocene hypothesis applicable to the temperate European region and that of the pre-Quaternary hypothesis in tropical Amazonian regions.

## MATERIALS AND METHODS

### Material collection

Two sets of taxa for different analyses were assembled. First, to examine the monophyly of each Formosan genus and species belonging to the subfamily Saturniinae, one nuclear gene of 28S rDNA and two mitochondrial genes of COI and 16S rDNA were applied, and two members in Salassinae, one subfamily belonging to Saturniidae, were applied as outgroups. A total of 52 individuals of seven genera and 18 taxa from various parts of Taiwan, Vietnam and China were analyzed (Table S1). Second, the barcoding initiative provides a multitude of Southeast and East Asian saturniine COI sequences, COI sequences in the first dataset and those belonging to the Formosan saturniine genera from South and East Asia were downloaded from GenBank and the Barcode of Life Data System (BOLD) to elucidate the speciation timing and possible origin of Formosan saturniines. The specific names of the analyzed saturniine species follow the recent review in *Kitching et al. (2018)*. Crucial information, collected in 519 sequences and the aligned sequences of 114 taxa is provided in Table S1 and Fig. S1. To prevent the misidentification

of the downloaded sequences, information based on the monophyly and distribution area of each saturniine species was employed to accurately retrieve sequences from GenBank.

Voucher specimens used only in the present study were stored at −20 °C at the Department of Entomology, National Chung Hsing University. A document permitting the collection of saturniine insects was acquired from the Yushan, Shei-Pa, Taroko, and Kinmen national parks in Taiwan.

## DNA extraction, amplification, and direct sequencing

One leg of each voucher specimen extracted for DNA was modified from *Yeh et al. (2004)* using the Wizard Genomic DNA Purification Kit (Promega, Madison, WI, USA). Crude DNA dissolved in 100 μl of ddH$_2$O was used as a template in subsequent polymerase chain reaction (PCR). Two mitochondrial genes, COI and 16S rDNA, and one nuclear gene, 28S rDNA, were amplified. K762UBombyx and Jerry D primers (*Caterino & Sperling, 1999*) (*i.e.*, barcode primers) were used to amplify and sequence the COI fragment. The primer sets used for amplification and sequences of 16S rDNA and 28S rDNA were 16SR21-16S22 and 28Sa-28Sb, respectively (*Yeh, Yang & Kang, 1997*; *Lin, Wang & Yeh, 2003*). For COI, 35 cycles of amplification were performed in a final volume of 50 μl containing 10 mM Tris-Cl (pH 9.0), 50 mM KCl, 1.5 mM MgCl$_2$, 0.01% gelatin, 0.1% Triton-X100, 0.2 mM of each dNTP, 10 pmoles of each primer, 2 μl of the DNA template, and 2 units of SuperTaq polymerase (Protech Technology, Taipei, Taiwan). The PCR assays for 16S rDNA and 28S rDNA were performed in a volume of 25 μl mixture containing 1 μl of template DNA, 2.5 μl of 10× Taq buffer, 0.2 μl of 25 mM dNTP, 0.5 μl of *Taq* DNA polymerase, and 0.5 μl of primers. The PCR programming conditions were 95 °C for 2 min as the first denaturation followed by 35 cycles of denaturation at 95 °C for 1 min, annealing at 42–52 °C for 1 min, and extension at 72 °C for 50–60 s, with a final extension at 72 °C for 10 min. The amplified DNA fragment was excised and purified using the QIAquick Gel Extraction Kit (Qiagen, Hilden, Germany) after resolution on agarose gel. The resultant DNA product was sequenced using a Taq dye terminator cycle sequencing kit (Applied Biosystems, Waltham, MA, USA) and ABI 377A sequencer. Sequences amplified in the present study have been deposited in GenBank under AB533547–AB533597 for the COI gene, AB841436–AB841480 for the 16S rDNA gene, and AB841488–AB841532 for the 28S rDNA gene (Table S1).

## DNA analysis

Sequences of saturniines were compiled using the BioEdit program (*Hall, 1999*) and aligned with pertinent sequences retrieved from GenBank and the BOLD before being manually checked (Fig. S1). Pairwise distance was estimated using an uncorrected proportional divergence in MEGA7 software (*Kumar, Stecher & Tamura, 2016*).

Phylogenetic inferences of Formosan saturniines were made using the maximum likelihood (ML) approach and Bayesian inference (BI). The ML tree was executed using the online IQ-TREE 1.6.11 (http://iqtree.cibiv.univie.ac.at/) (*Nguyen et al., 2015*). The Edge-unlinked partition model was selected for the partition type because each amplicon has its own branch length. The best substitution model was selected using the "Auto" option and

the FreeRate heterogeneity (+R). One hundred bootstrap replicates were used to test the statistical support of each node. MrBayes v.3.2 (*Ronquist et al., 2012*) was used to conduct BI, and the substitution model was estimated using jModelTest 0.1 (*Posada, 2008*). The evolutionary hypotheses for the best-fit substitution models of COI, 16S rDNA, and 28S rDNA were TIM1+I+G, TPM2uf+G, and TPM2+I+G, respectively, according to the Bayesian information criterion. The BIs of the three partitioned genes were analyzed with three heat chains and one cold chain, and Markov chain Monte Carlo searches were conducted for $10^7$ generations with sampling every 100 generations. The analysis was based on an average standard deviation of split frequencies lower than 0.01. The initial 25% of trees were discarded as burn-in, and the remaining trees were used to generate a consensus tree. Moreover, all the relevant Southeast and East Asian saturniine COI downloaded from GenBank were used to construct a ML tree using IQ-TREE 1.6.11 (http://iqtree.cibiv.univie.ac.at/) (*Nguyen et al., 2015*). The best substitution model was selected using the "Auto" option and the FreeRate heterogeneity (+R). One thousand Ultrafast bootstrap analyses were used to assess the statistical support of each node. Sibling species and those closely related to Formosan saturniine species were analyzed based on the reconstructed COI tree.

The haplotype network of each Formosan saturniine COI sequence was analyzed based on sibling or closely related saturniine species by using the TCS program with a 95% connection limitation (*Clement, Posada & Crandall, 2000*).

Calibration dating for the coalescent time of the separation between Formosan saturniines and their mainland or sibling saturniine lineages was estimated using a strict molecular clock in BEAST v.1.6.1 (*Drummond & Rambaut, 2007*). Different substitution rates for COI applied in insects were employed to estimate the relative divergent time since no fossil record has been recorded for Formosan saturniines and their sibling lineages (*Sohn et al., 2012*). Two substitution rates for COI applied in Lepidoptera were employed. A substitution rate of 1.15% per lineage in 1 million year for COI was used because it has been calibrated by the geological event and optimally applied for closely related taxa (*Brower, 1994*); While the substitution rate of 1.55% proposed by *Nazari & Sperling (2007)* was applied as well. Markov chain Monte Carlo sampling was conducted on $10^8$ generations with sampling every $10^4$ generations, depending on the effective sample size of the parameter estimated using Tracer software v.1.5 for each saturniine species (*Rambaut & Drummond, 2009*). The trees were recorded every 10,000 generations, and 10% of the recorded trees were burn-in.

## RESULTS

### Sequence composition of COI, 16S rDNA, and 28S rDNA in Saturniinae species

COI, 16S rDNA, and 28S rDNA were successfully amplified for 52 individuals of 18 taxa in 7 genera, with fragment sizes of 643, 486, and 654 bp, respectively. The downloaded sequences of COI and 16S rDNA from GenBank were 467 and 43, respectively. The average base compositions, including the downloaded sequences for G, A, T, and C,

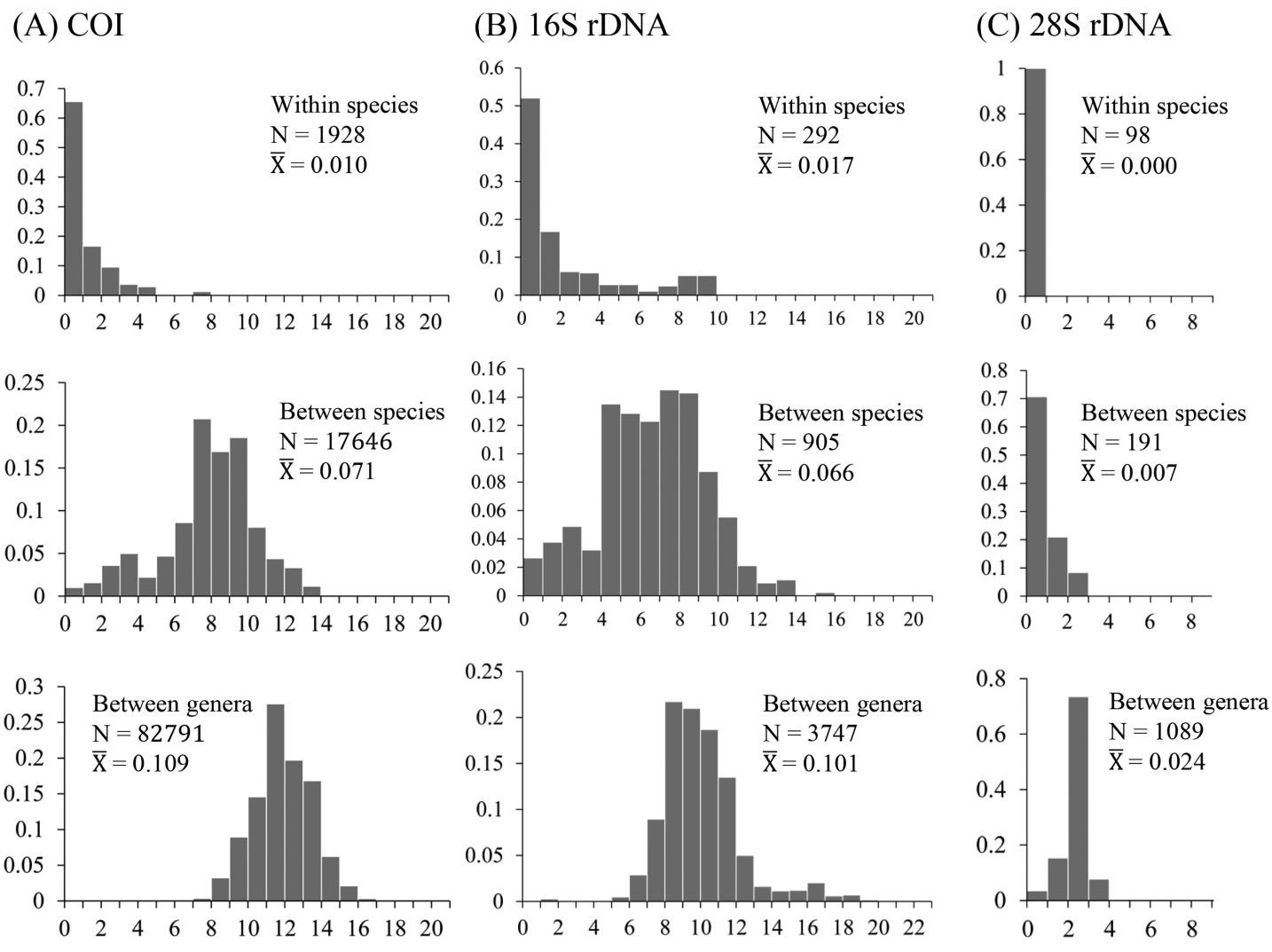

**Figure 1 Frequency distribution of p-distance divergence in sequences of COI (A), 16S rDNA (B), and 28S rDNA (C) within different taxonomic categories.** The number of pairwise comparisons (N) and the average divergence of each category are presented. Sequence divergences and their proportions are indicated on the X- and Y-axes, respectively.

are 14.9%, 28.8%, 39.5%, and 16.8% in the COI gene; 13.9%, 39.8%, 38.7%, and 7.6% in 16S rDNA; and 30.6%, 25.8%, 20.8%, and 22.8% in 28S rDNA; respectively.

## Sequence variations in Saturniinae species

The uncorrected nucleotide divergence and distribution frequencies in Saturniinae for each COI, 16S rDNA, and 28S rDNA in three categories were as follows: (1) 0–8%, 0–10%, and 0–1%, respectively, among individuals within species; (2) generally 6–11%, 0–11%, and 0–3%, respectively, among species of a given genus; and (3) generally 10–15%, 8–13%, and 2–3%, respectively, among genera (Fig. 1). Most saturniine species have highly similar COI and 16S rDNA sequences within species, with intraspecific divergence generally being <2%, whereas the interspecific and intergeneric genetic divergence was

6–12% (Figs. 1A and 1B). The divergence of the conserved 28S rDNA was generally low (<2%), and for many congeneric species, no genetic variations were detected (Fig. 1C).

Approximately 90% of uncorrected nucleotide divergences among COI within species were <2% (Fig. 1A); all divergences exceeding 2% originate from comparisons between allopatric populations or subspecies. For example, a considerable divergence of 4.8% was discovered in the population of *Att. atlas* in Taiwan from other allopatric populations of *Att. atlas* in Indonesia and Vietnam. By contrast, subtle or few genetic divergences were noted in allopatric species or subspecies (data provided below). The Taiwanese subspecies *Saturnia japonica arisana* (Shiraki) has an identical sequence to its allopatric nominate subspecies *Saturnia japonica japonica* (Moore) from China (data provided below).

Genetic divergences >0.5% among individuals have been observed in several Taiwanese lineages, such as *Loepa mirandula* Yen, Nässig, Naumann & Brechlin, *Actias ninpoana ningtaiwana* Brechlin, *Att. atlas*, *Saturnia pyretorum* Westwood, and *Samia wangi* Naumann & Peigler (data provided below).

## Phylogenetic inferences of Saturniinae species

Phylogenetic inferences based on COI + 16S rDNA + 28S rDNA using BI and ML methods revealed that the sampled Saturniinae formed a monophyletic group comprising monophyletic genera and species (Fig. 2); this provides essential information to document the evolutionary differentiation events between sibling or closely related saturniines. Moreover, 519 COI sequences in 114 saturniine species related to Taiwanese *Actias*, *Antheraea*, *Attacus*, *Loepa*, *Rhodinia*, *Samia*, and *Saturnia* were downloaded from GenBank to analyze the differentiation history of each Taiwanese saturniine (Fig. S1, Table S1). The phylogenetic topology (Fig. S2) revealed that these Asian saturniine species are primarily monophyletic, although some of them exhibited paraphyletic deduced allopatric closely related species. For example, the endemic *Loepa formosensis* Mell and *Rhodinia verecunda* Inoue in Taiwan form terminal lineages within the widespread Southeast Asian *Loepa diffunorientalis* Brechlin and East Asian *Rhodinia fugax* Butler, respectively.

## Phylogenetic relationships between insular Taiwanese saturniines and their Asian allies

On the basis of COI phylogenetic relationships (Fig. S2), the potential sister saturniine species, subspecies, and populations related to Taiwanese saturniines were used to estimate the differentiation and divergence events of each Taiwanese saturniine. However, no species closely related to *Actias neidhoeferi* Ong & Yu was discovered (Fig. S2).

## Differentiation in the species category of Taiwanese saturniines

All six saturniine species endemic to Taiwan were found to be monophyetic. Among them, *Loe. mirandula* exhibited the greatest divergence from its sibling taxa and *Rho. verecunda* exhibited the least divergence. In *Loe. mirandula*, the network analysis revealed >30 substitution steps to its possibly sibling *Loepa miranda* Moore (Yen et al., 2000) and the

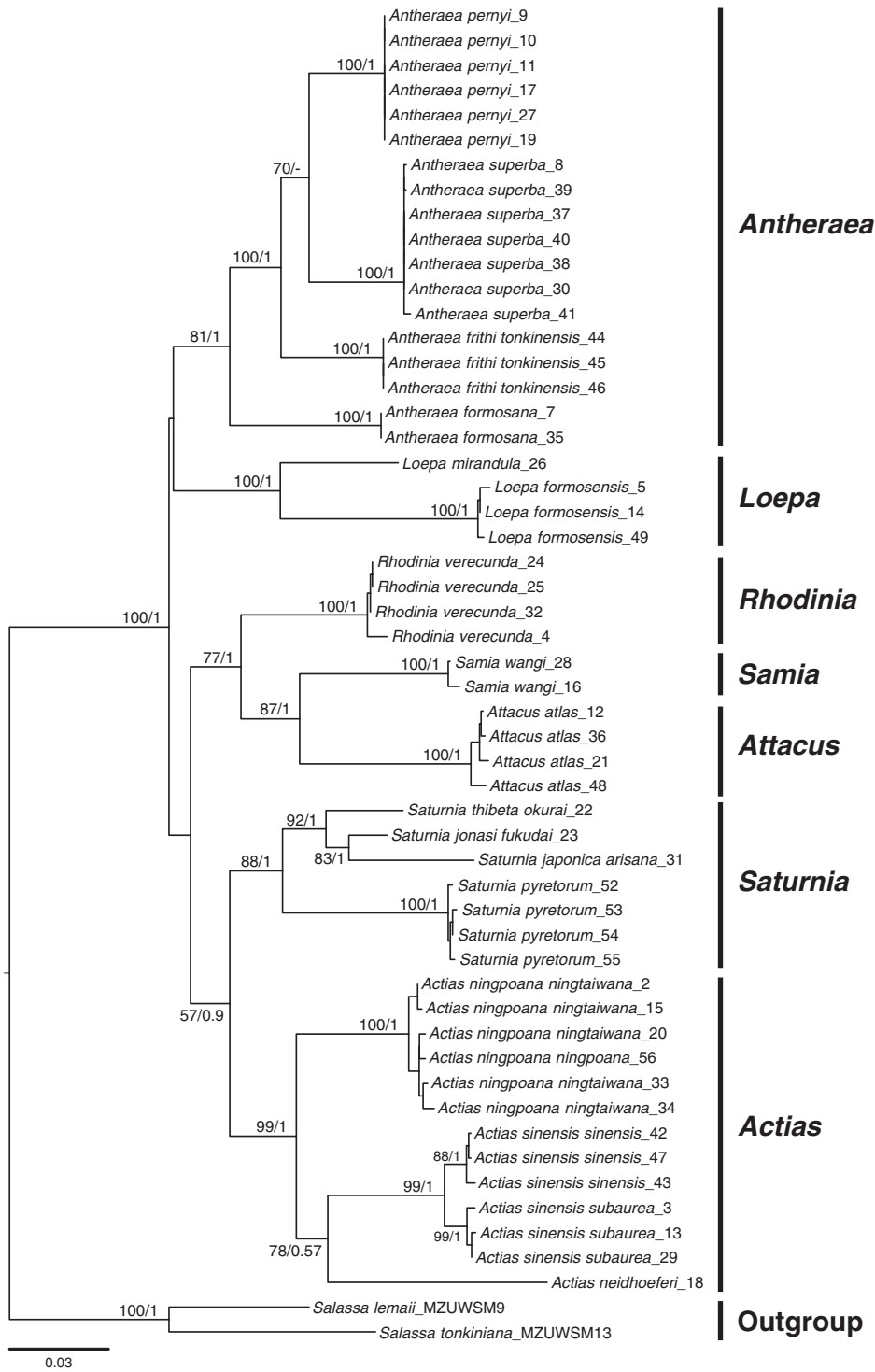

**Figure 2 Phylogenetic tree constructed from COI + 16S rDNA + 28S rDNA sequences using the maximum likelihood (ML) approach and Bayesian inference for Formosan Saturniidae.** Bootstrap values of the ML tree and the posterior possibility from the Bayesian analysis are provided beneath the node. Two outgroups used were from saturniid subfamilies of Salassinae.

## *Loepa mirandula*

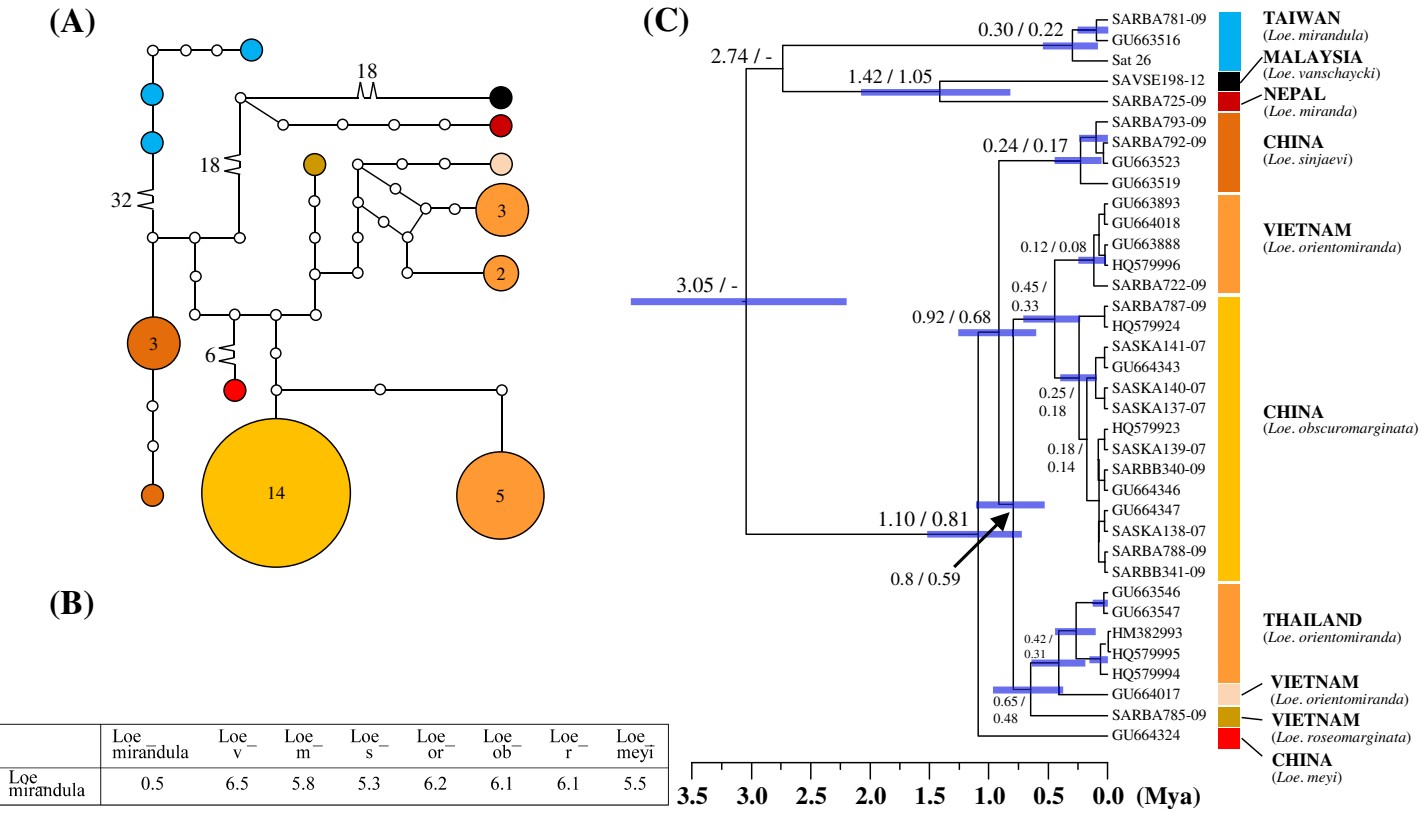

**Figure 3 Haplotype network analysis (A), sequence divergence (B), and calibration dating (C) of COI sequences for the Formosan endemic *Loepa mirandula* and its ally species.** The samples and geographical distributions of all *Loepa* species are indicated by different colors. The abbreviations Loe_v, Loe_m, Loe_s, Loe_or, Loe_ob, Loe_r, and Loe_meyi refer to *Loe. vanschaycki*, *Loe. miranda*, *Loe. sinjaevi*, *Loe. orientomiranda*, *Loe. obscuromarginata*, *Loe. roseomarginata*, and *Loe. meyi*, respectively. In the haplotype network (A), each circle represents a haplotype connected to another haplotype through one substitution step, and the numbers of substitution steps greater than one are marked. The number of haplotype individuals greater than one is marked inside the circle, with the smallest circle corresponding to one individual. Sequence divergences (B) in pairwise taxa are presented as percentages (%). Calibration dating is provided in millions of years ago (Mya) (C).

closely related *Loepa vanschaychi* Brechlin, *Loepa sinjaevi* Brechlin, *Loepa orientomiranda* Brechlin & Kitching, *Loepa obscuromarginata* Naumann, *Loepa roseomarginata* Brechlin, and *Loepa meyi* Naumann (Fig. 3A). Three haplotypes with a maximum of five substitution steps were discovered in *Loe. mirandula*. For *Loe. mirandula*, the intraspecific sequence divergence was 0.5% on average, and the divergence from its allied Asian species was 5.3–6.5% (Fig. 3B). The differentiation event of *Loe. mirandula* from its sibling saturniine species was estimated to have occurred approximately 2.7 Mya (Fig. 3C).

For Taiwanese *Rho. verecunda*, two haplotypes with four substitution steps were found, differentiating *Rho. fugax* into two haplogroups (Fig. S3). Average sequence divergence within *Rho. verecunda* was 0.3%, and that from *Rho. fugax* and *Rhodinia newara* (Moore) was 0.7% and 3.7%, respectively (Fig. S3B). Network and phylogenetic analyses revealed that the Taiwanese endemic *Rho. verecunda* produces *Rho. fugax* paraphyly. The splitting events of *Rho. verecunda* and one of the *Rho. fugax* lineage occurred approximately 0.23 Mya (Fig. S3).

For Taiwanese *Loe. formosensis*, a network analysis revealed four haplotypes with a maximum of five substitution steps and >17 substitution steps to the closely related *Loe. diffunorientalis* distributed in Vietnam (Fig. S4A). On average, intraspecific sequence divergence was 0.4% within *Loe. formosensis* and 3.5–5.3% from its related saturniines *Loe. diffunorientalis*, *Loepa diffunoccidentalis* Brechlin, and *Loepa nepalensis* Brechlin (Fig. S4B). A speciation event of *Loe. formosensis* and its sibling *Loe. diffunorientalis* was estimated to have occurred approximately 1.5 Mya (Fig. S4C).

A network analysis revealed no variation in the Taiwanese species of *Antheraea superba* Inoue but substantial sequence divergence of 3.8% from its sister species *Antheraea yamamai* Guérin-Méneville from Japan and Korea and that from China (Fig. S5). The differentiation event of the two species can be dated back to approximately 1.3 Mya (Fig. S5C).

The network analysis also indicated that the Taiwanese *Antheraea formosana* Sonan groups with Indian *Antheraea assamensis* (Helfer) were related to *Antheraea paniki* Nässig & Treadaway from the Philippines with >16 substitution steps (Fig. S6A). *Antheraea formosana* was noted to exhibit an average sequence divergence of 1.1% and 3.4% from *Ant. assamensis* and *Ant. paniki*, respectively (Fig. S6B). The speciation events for these three saturniine species seems to have occurred <1 Mya, whereas those for *Ant. formosana* and Indian *Ant. assamensis* occurred <0.5 Mya (Fig. S6C).

Network analysis for Taiwanese endemic *Saturnia fukudai* Sonan revealed 23 substitution steps to its sibling lineage of *Saturnia boisduvalii* Eversmann from Korea (Fig. S7A), reflecting a sequence divergence of 4% between the two (Fig. S7B). Lineage calibration revealed that the estimated divergence was approximately 1.3 Mya between them (Fig. S7C).

## Differentiation in the subspecies category of Taiwanese saturniines

Three of four Taiwanese endemic subspecific saturniines were monophyletic. Moreover, a genetic admixture caused *Sat. jap. japonica* to be paraphyletic. *Actias ningpoana* (C. & R. Felder), commonly known as the Chinese luna moth, distributes throughout East China and the islands of Taiwan and Hainan. The Taiwanese endemic subspecies *Act. nin. ningtaiwana* comprises two genetic lineages, one of which (lineage II) has a close affinity to individuals of the subspecies *Actias ningpoana ningpoana* (C. & R. Felder) from China (Fig. 4). The genetic admixture in the two subspecies led to high intrasubspecific sequence divergence within the subspecies *Actias nin. ningtaiwana* of up to 0.7% (Fig. 4B). The calibration time revealed that the divergence between Taiwanese lineages I and II occurred approximately 0.45 Mya (Fig. 4C). Taiwanese lineage I can be found only in Taiwan, whereas lineage II has an admixture relationship with the Chinese population, which split approximately 0.1 Mya.

Two *Saturnia* species from Taiwan were recognized as endemic subspecies. Network analysis for Taiwanese endemic subspecies of *Saturnia thibeta okurai* (Okano) revealed approximately 23 substitution steps to its neighboring subspecies of *Saturnia thibeta pahangensis* Brechlin and *Saturnia thibeta extensa* Butler from Malaysia and India, respectively (Fig. S8A). Sequence divergences between the Taiwanese and other subspecies

## *Actias ningpoana ningtaiwana*

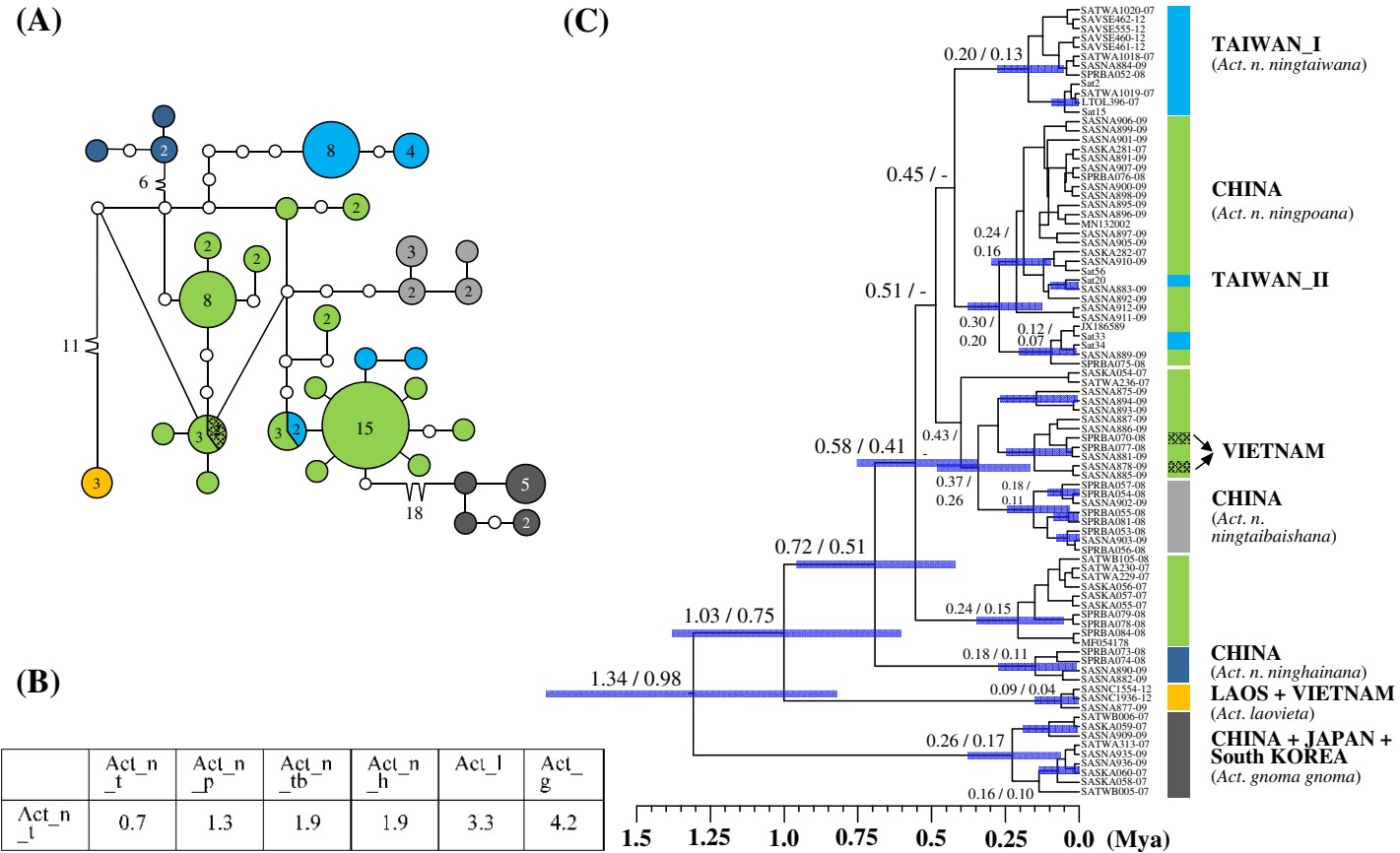

**(A)**

**(C)**

**(B)**

| | Act_n_t | Act_n_p | Act_n_tb | Act_n_h | Act_l | Act_g |
|---|---|---|---|---|---|---|
| Act_n_t | 0.7 | 1.3 | 1.9 | 1.9 | 3.3 | 4.2 |

**Figure 4** **Haplotype network analysis (A), sequence divergence (B), and calibration dating (C) of COI sequences for *Actias ningpoana*** **subspecies and its ally *Actias* species.** The samples and geographical distributions of all relevant subspecies are indicated by different colors. The abbreviations Act_n_t, Act_n_p, Act_n_tb, Act_n_h, Act_l, and Act_g refer to *Act. nin. ningtaiwana*, *Act. nin. ningpoana*, *Act. ningtaibaishana*, *Act. ninghainana*, *Act. laovieta*, and *Act. gnoma gnoma*, respectively. In the haplotype network (A), each circle represents a haplotype connected to another haplotype through one substitution step, and the numbers of substitution steps greater than one are marked. The number of haplotype individuals greater than one is marked inside the circle, with the smallest circle corresponding to one individual. Sequence divergences (B) in pairwise taxa are presented as percentages (%). Calibration dating is provided in millions of years ago (Mya) (C).

were approximately 4% (Fig. S8B), and lineage calibration revealed a divergence of approximately 1.7 Mya between them (Fig. S8C).

For the Taiwanese endemic subspecies of *Sat. jap. arisana*, an identical haplotype has been found in several mainland *Sat. jap. japonica* individuals, indicating a recent migration event of this Taiwanese subspecies (Fig. S9A). Moreover, the lineage *Sat. jap. arisana* was found to have a paraphyletic relationship with *Sat. jap. japonica* (Fig. S9C). In addition, the complete mitochondrial genome of *Saturnia boisduvalii* (Eversmann) (Accession No. EF622227) might be a misidentification of *Sat. jap. japonica* (Fig. S9C).

In the Taiwanese endemic *Act. sin. subaurea*, three haplotypes with a maximum of three steps were discovered; however, only one haplotype existed in its sibling *Actias sinensis sinensis* (Walker) from China and Vietnam (Fig. S10A). The haplotype network revealed that both *Act. sin. subaurea* and *Act. sin. sinensis* have more than 28 substitution steps

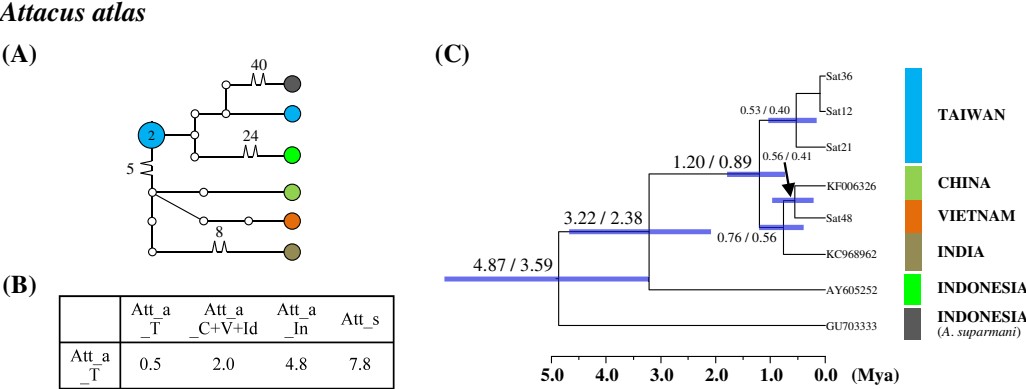

*Attacus atlas*

**(A)**

**(B)**

|  | Att_a _T | Att_a _C+V+Id | Att_a _In | Att_s |
|---|---|---|---|---|
| Att_a _T | 0.5 | 2.0 | 4.8 | 7.8 |

**(C)**

TAIWAN
CHINA
VIETNAM
INDIA
INDONESIA
INDONESIA (*A. suparmani*)

**Figure 5** **Haplotype network analysis (A), sequence divergence (B), and calibration dating (C) of COI sequences for the Formosan *Attacus atlas* population and its allopatric populations.** The samples and geographical distributions of all *Att. atlas* allopatric populations are indicated by different colors. The abbreviations Att_a_T, Att_a_C+V+Id, and Att_a_In refer to populations from Taiwan, China +Vietnam+India, and Indonesia, respectively, and Att_s is *Attacus suparmani*. In the haplotype network (A), each circle represents a haplotype connected to another haplotype through one substitution step, and the numbers of substitution steps greater than one are marked. The number of haplotype individuals greater than one is marked inside the circle, with the smallest circle corresponding to one individual. Sequence divergences (B) in pairwise taxa are presented as percentages (%). Calibration dating is provided in millions of years ago (Mya) (C).

related to their potential sister species *Actias parasinensis* Brechlin from Bhutan (Fig. S10A). A distinct genetic divergence of 2.1% was noted between *Act. sin. subaurea* and *Act. sin. sinensis* (Fig. S10B), with a split event at approximately 0.9 Mya according to lineage calibration (Fig. S10C).

## Differentiation in the Taiwanese allopatric population

Population differentiation analysis revealed that all Taiwanese saturniine populations belonged to monotypic species. A recent migration event for Taiwanese populations would thus be expected. Additionally, substantial divergence was found in the Taiwanese *Att. atlas* population. The attractive giant monotypic *Att. atlas* distributed widely throughout tropical regions, from India eastward to the Philippines and Taiwan. The network indicated that two divergent haplotypes exist in the Taiwanese *Att. atlas*, with sequence divergences of 2% and 4.8% from the China–Vietnam–India region and Indonesia, respectively (Fig. 5). The allopatric *Att. atlas* populations from the three regions formed distinct lineages (Fig. 5C). Lineage calibration estimated that the speciation event of *Att. atlas* occurred approximately 2.5 Mya. The Taiwanese lineage diverged from the China–Vietnam–India lineage approximately 1.2 Mya.

The monotypic *Antheraea pernyi* (Guérin-Méneville), which has seven haplotypes in Taiwan, China, and Korea, exhibits average sequence divergences of 1.7%, 2.4%, and 2.6% from the allied species *Antheraea vietnamensis* Brechlin & Paukstadt, *Antheraea jawabaratensis* Brechlin & Paukstadt, and *Antheraea roylei* Moore, respectively (Fig. S11). The differentiation event of the Taiwanese *Ant. pernyi* population could be dated back to approximately 0.4 Mya (Fig. S11C). Calibration dating indicated high affinity among *Ant. pernyi*, *Ant. jawabaratensis*, *Ant. vietnamensis*, and *Ant. roylei* and with

differentiation <1.4 Mya. However, the molecular identification of *Ant. pernyi* and its allies, such as *Ant. roylei* and *Ant. vietnamensis*, have been complicated by the domesticated hybrids of these species that were used for silk production (*Nagaraju & Jolly, 1986*; *Peigler & Naumann, 2003*).

*Saturnia pyretorum* Westwood distributed in Taiwan and China and has two distinct lineages, each of which has divergent individuals (Fig. S12). Sequence divergence between the two lineages was 1.4–2.2%. Calibration dating revealed two split events between Taiwanese and mainland lineages, approximately 0.25 and 0.1 Mya, respectively, indicating multiple migrations of *Sat. pyretorum* to Taiwan.

*Samia wangi* Naumann & Peigler is a monotypic species distributed in southern China and Taiwan. The shared haplotype between the Taiwan and mainland China populations indicated that a dispersal event may have occurred recently (Fig. S13). Calibration dating also revealed a recent migration event. The sequence divergences of *Sam. wangi* to the two *Samia* species of *Samia cynthia* (Drury) and *Samia ricini* (Jones) were approximately 1.9% and 3.8%, respectively (Fig. S13).

## DISCUSSION

### Sequence divergence for insular saturniines to their allopatric populations and closely related alliances

The present molecular study indicates that the taxonomy of several saturniine species endemic to Taiwan should be reassessed. Species recognition would be indefinite if the analyzed species were under speciation processes. Moreover, recognizing one differentiation population as a subspecies within a species or as a distinct full species is also challenging. In the past two decades, DNA barcodes have provided much data, revealing that a 2% COI divergence can be usefully applied to identify species. In the current study, phylogenetic topology revealed that the analyzed saturniine species were primarily monophyletic, with some being paraphyletic (Fig. S2). Several Formosan saturniine lineages exhibited COI divergences >2% from their sibling lineages, whereas others had divergences <2%. To document the taxonomic debates pertaining to Taiwanese saturniines, we followed the recent taxonomic review of Saturniidae by *Kitching et al. (2018)*. The frequencies of sequence divergences of each Taiwanese saturniine species, subspecies, and population from its closely related alliances are presented together in Fig. 6.

Sequence divergences within Formosan endemic species were <0.8% and >0.8% with their closely related alliances, except for *Rho. verecunda* (Fig. 6A). *Anteraea superba* has been considered a subspecies of *Ant. yamamai*, commonly known as the Japanese oak silkmoth, is a temperate saturniine distributed primarily throughout northern China, Korea, and Japan (*Peigler & Wang, 1996*; *Zhu & Wang, 1996*). These two species have highly similar morphological characteristics, although the wing color pattern varies from bright yellow to reddish brown among *Ant. yamamai* populations (*Peigler & Wang, 1996*). A substantial sequence divergence (>3.4%) from the sister species of *Ant. yamamai* indicates that *Ant. superba* in Taiwan is a separate evolutionary unit, either recognized in

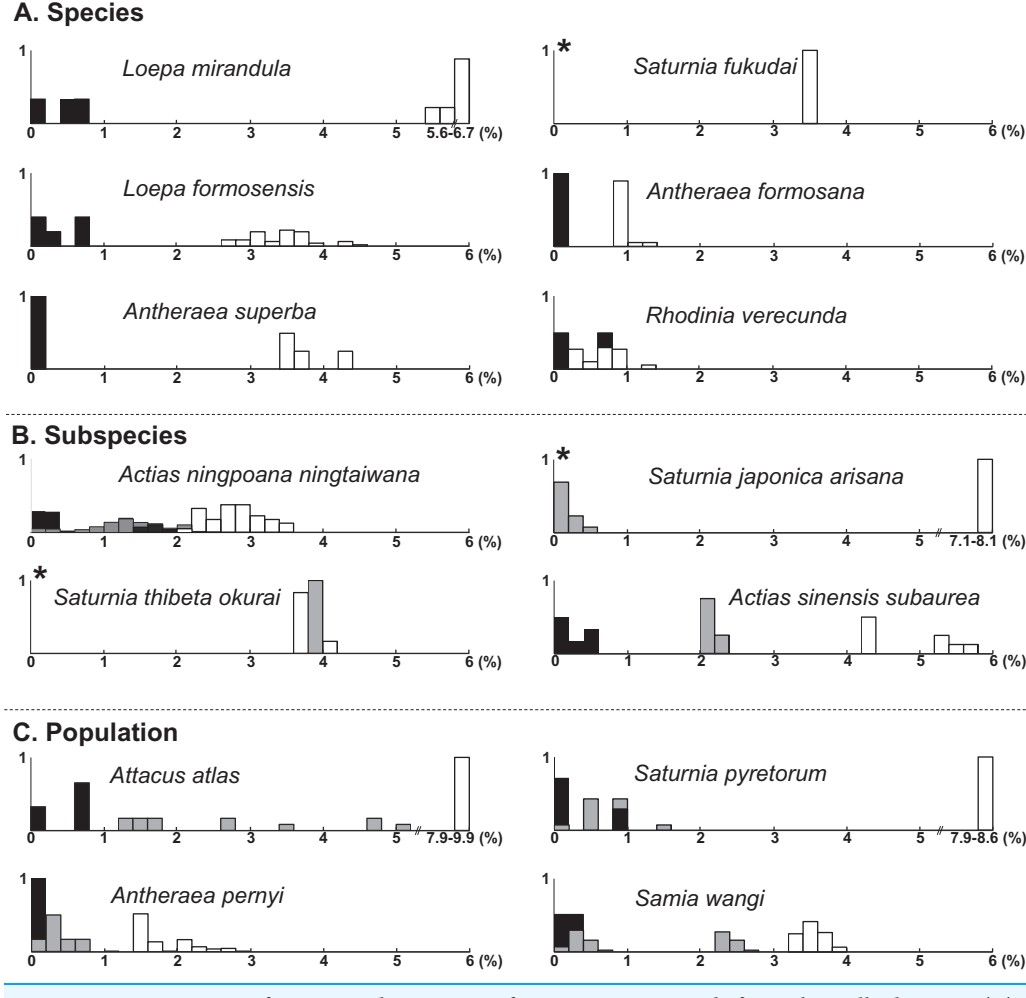

**Figure 6  A summary of sequence divergences of Formosan saturniids from their allied species (A), subspecies (B), and populations (C) distributed in East and South Asia.** The symbols of empty, gray, and black columns present sequence divergences within Formosan saturniids, between population or subspecies in a given species, and between sibling species, respectively. An asterisk (*) indicates no pairwise comparison within the Formosan saturniid population.

species or subspecies categories (Fig. 6A). Formosan *Sat. fukudai* has been considered an endemic subspecies of *Sat. jonasii* (*Peigler & Wang, 1996*; *Zhu & Wang, 1996*), and *Kitching et al. (2018)* treated it as a distinct species. Sequence divergences of approximately 4% in *Sat. fukudai* from its sibling species, *Sat. jonasii* and *Sat. boisduvalii*, suggest that *Sat. fukudai* in Taiwan is a separate evolutionary unit (Fig. 6A). *Rhoinia verecunda*, endemic to Taiwan, produced a paraphyletic *Rho. fugax*, involving several subspecies in China (Fig. S3). Intraspecific sequence divergences within *Rho. verecunda* were 0–0.8% and between *Rho. verecunda* and *Rho. fugax* were 0.2–1.4% (Fig. 6A); however, the color patterns and wing bandings between them could be distinguished, thereby indicating inconsistencies between morphological characteristics and molecular divergence.

Sequence divergences were <2% within Formosan subspecies, >2.2% from their closely related species, and 0–4% from their allied subspecies (Fig. 6B). *Actias nin. ningtaiwana*,

which is endemic to Taiwan, is known subspecies of *Actias selene ningpoana* C. & R. Felder of *Actias selene* (Hübner), more commonly known as the Indian luna moth, which is distributed throughout South Asia (*Peigler & Wang, 1996*). *Brechlin (2012)* highlighted that several *Act. ningpoana* subspecies were distinct from *Act. selene*. The present study identified sequence divergences larger than 2.2% (Fig. 6B) and a splitting time between *Act. ningpoana* and *Act. laovieta* Naumann, Nässig & Löffler of approximately 1.0 Mya (Fig. 4). Genetic variations with an overlapping of sequence divergences between intersubspecies and intrasubspecies were observed: 0–2% within *Act. nin. ningtaiwana*, 0–2% between *Act. nin. ningtaiwana* and *Act. nin. ningpoana*, and 0.7–2.1% between *Act. nin. ningpoana* and *Act. nin. ningtaibaishana*; therefore, their subspecific nomenclature was open to debate (Fig. 6B). For *Act. sin. subaurea*, *Zhu & Wang (1996)* suggested that *Act. sin. subaurea* should be a species distinct from *Act. sinensis* and be called *Actias heterogyna* (Mell). The present study revealed that a distinct genetic divergence of 2–2.4% existed between *Act. sin. subaurea* and *Act. sin. sinensis*, with a split event approximately 0.9 Mya (Fig. S10). Therefore, *Act. sin. subaurea* in Taiwan is a separate evolutionary unit that is either recognizable as a species or subspecies category (Fig. S10). Wing color and pattern are similar among *Sat. thibeta* subspecies, although the wing bandings are more pointed in Formosan *Sat. thi. okurai* (*Roepke, 1953*; *Pinratana & Lampe, 1990*; *Peigler & Wang, 1996*; *Zhu & Wang, 1996*). Sequence divergences of approximately 4% in *Sat. thi. okurai* to its related subspecies of *Sat. thi. pathangensis* may raise the issue of species splitting (Fig. 6B). A haplotype of a Formosan subspecies of *Sat. jap. arisana* identical to some mainland *Sat. jap. japonica* individuals shows no variation (Fig. 6B), although wing banding is clear and thicker in *Sat. jap. arisana* (*Roepke, 1953*; *Pinratana & Lampe, 1990*; *Peigler & Wang, 1996*; *Zhu & Wang, 1996*).

At the population level, overlapping sequence divergences were generally found within and between populations, which might be caused by recent dispersal or multiple migration events such as those for *Sat. pyretorum* and *Sam. wangi* (Fig. 6C); no taxonomic debate has been proposed except for one concerning in *Att. atlas*. The attractive giant *Att. atlas* is distributed widely throughout tropical regions, from India eastward to the Philippines and Taiwan, and several subspecies of varying size, coloration, and wing pattern have been proposed (*Roepke, 1953*; *Pinratana & Lampe, 1990*; *Peigler & Wang, 1996*; *Zhu & Wang, 1996*). However, *Kitching et al. (2018)* suggested that the proposed *Att. atlas* subspecies should be considered synonymous with *Att. atlas*. Sequence divergences with ranges of 1.2–5.2% among several allopatric *Att. atlas* populations suggest distinct evolutionary units among them (Fig. 6C). For the Formosan *Sat. pyretorum*, two distinct genetic lineages were likely to have multiple dispersal events that produces the genetic admixture between insular and mainland populations (Fig. 6C, Fig. S12); however, *Sat. pyretorum* was artificially introduced to Taiwan in the first half of the 20th century (*Chu, 2005*). Accordingly, the divergence between the natural Taiwanese population and the mainland population should be 0.8–1.4%. No sequence variation was found in some individuals of *Sam. wangi* from Taiwan and the mainland, implying a recent migration event (Fig. 6C).

## Calibration dating and origin of island saturniines

No fossil records have been reported for Taiwanese saturniines and their close affinities to infer their absolute divergent time. Molecular calibration based on COI sequences for East and Southeast Asian saturniines can help in determine the spatiotemporal patterns of insular saturniines. Speculation suggests that organisms in Taiwan primarily originated from neighboring mainland China, Southeast Asia, and Northeast Asia, either through currents or the land bridges formed during glaciation (*Voris, 2000*; *Ray & Adams, 2001*; *Chang & Chen, 2005*; *Jang-Liaw, Lee & Chou, 2008*; *Huang & Lin, 2010*; *Tsai, Wan & Yeh, 2014*; *Yu, Lin & Weng, 2014*; *Tsai & Yeh, 2016*; *Weng, Yang & Yeh, 2016*; *Kurita et al., 2017*). However, phylogenetic analyses indicated that most Formosan saturniines were distinct and monophyletic. Therefore, migration through currents across the Taiwan Strait might not be the most common route of saturniines' arrival to Taiwan.

Figure 7 present the split times accompanied by the geographical distribution of each Formosan saturniine species, subspecies, and population from its neighboring closely related alliances with Pleistocene epochs. In six Taiwan endemic saturniine species, calibration dating yielded sporadic splitting time within the range of 0.2–2.7 Mya (Fig. 7A). *Loepa mirandula*, *Loe. formosensis*, *Ant. superba*, and *Sat. fukudai* exhibited split events of >1.3 Mya (Early Pleistocene) and <0.5 Mya (Middle Pleistocene) in *Ant. formosana* and *Rho. verecunda*. Furthermoer, two subspecific endemic saturniines in Taiwan had splitting events 0.9–1.7 Mya (Early Pleistocene) and another two, *Sat. ningpoana ningtaiwana* and *Sat. japonica arisana*, might have had splitting events in the Middle and Late Pleistocene (Fig. 7B). However, the estimated divergence time of the subspecific Formosan *Sat. thi. okurai* might be <1.71 Mya when including mainland populations (Fig. S8); this is based on the fact that *Sat. thibeta* also exists in northern Vietnam and northern China (*Peigler & Naumann, 2003*). At the population level, the splitting time of Formosan populations from their mainland populations was estimated to be <0.4 Mya in the Middle and Late Pleistocene, except for *Att. atlas*, which had a splitting time of approximately 1.2 Mya (Early Pleistocene) (Fig. 7C). *Kawahara et al. (2019)* used phylogenomics to determine the evolutionary timing and pattern of butterflies and moths including *Actias*, *Saturnia*, and *Attacus* genera of Saturniinae. Phylogenomic analysis also revealed the sister relationships between *Actias* and *Saturnia* (Fig. 2). Figure 7A also indicates that the splitting time, 0.2–2.7 Mya, in species differentiation events was far less than the generic divergent time, 22–30 Mya, as proposed by *Kawahara et al. (2019)*.

*Merckx et al. (2015)* indicated that the montane adaptive biota in Mount Kinabalu, at an elevation of 4,095 m, on the tropical island Sabah mostly originated from long-range dispersal from cool localities elsewhere. The present study indicates that some insular saturniines might have originated from temperate regions in Northeast Asia, whereas others may have originated from southern China and Southeast Asia (Fig. 7). Taiwanese saturniines, including *Ant. superba*, *Sat. japonica arisana*, *Sat. fukudai*, and *Ant. pernyi*, predominantly inhabited montane or subalpine regions in Taiwan at altitudes >1,500 m and have related lineages or sibling species distributed mostly in temperate areas such as northern China, Japan, and Korea (*d'Abrera, 1998*). Taiwanese saturniines distributed

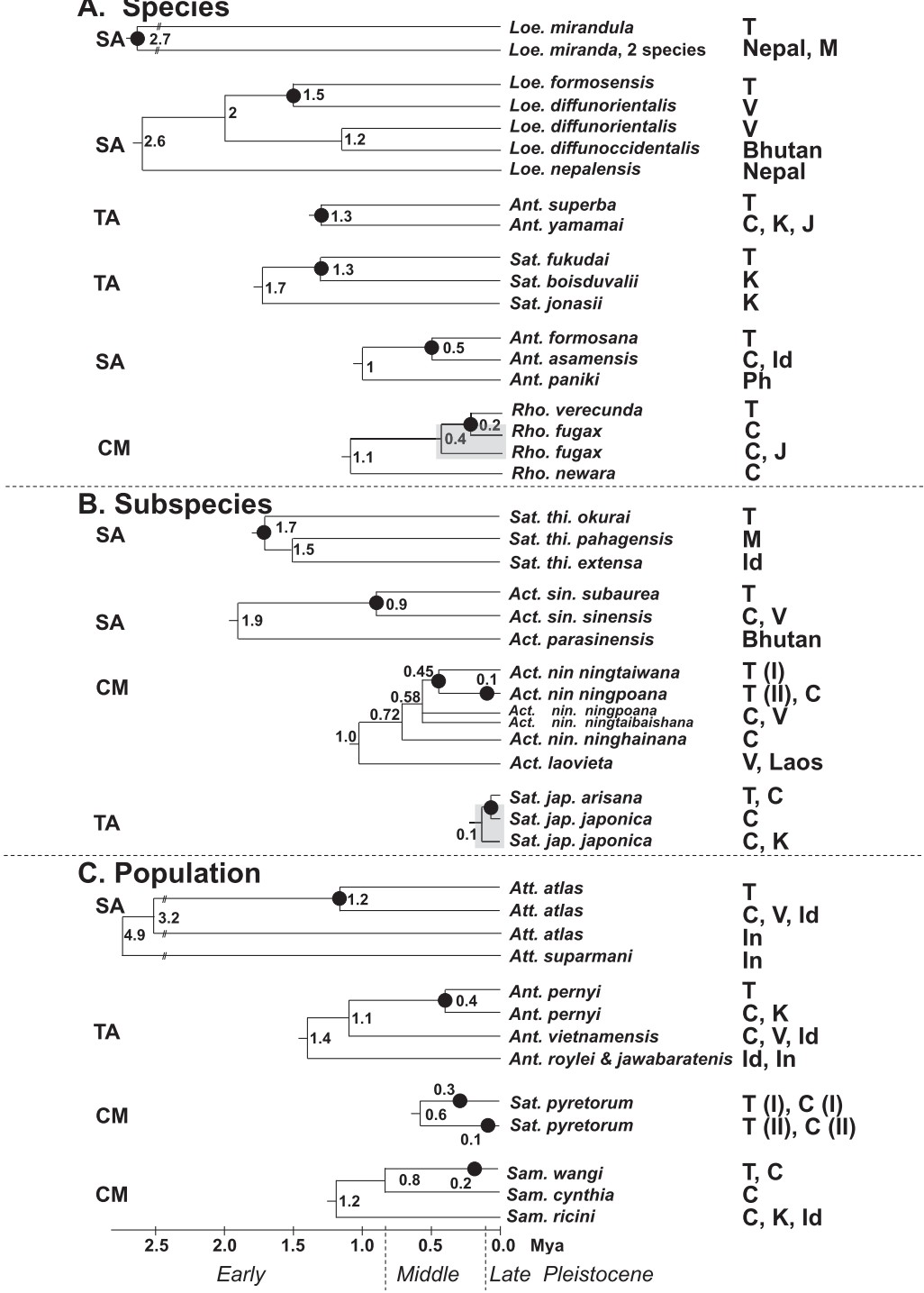

**Figure 7 Summary of the splitting time of Formosan saturniids from their close ally species (A), subspecies (B), and populations (C) distributed in East and South Asia.** The splitting event of the Formosan lineage from each relevant saturniine lineage is indicated by a dot (●). Relevant calibration dating (million years ago, Mya) is provided at the bottom and interior node. The geological boundaries in 0.8 and 0.12 Mya by vertical dot line are shown for the Pleistocene epoch. The paraphyletic saturniid

Figure 7 (continued)
taxon induced by the Taiwanese taxon is indicated by a gray rectangle. The capital SA, TA, and CM in front of each saturniid dendrogram denote Southeast Asia, temperate Northeast Asia, and the Chinese mainland, respectively, indicating the most likely relationships of each Formosan lineage. The abbreviations of country are as follows: China (C), India (Id), Indonesia (In), Japan (J), Korea (K), Malaysia (M), Philippines (Ph), Taiwan (T), and Vietnam (V). 

throughout plains and hills might not be related to temperate lineages. All species, subspecies, and populations of the Taiwanese *Act. sinensis*, *Ant. formosana*, *Att atlas*, *Loe. formosensis*, *Loe. mirandula*, and *Sat. thibeta* have closely related lineages distributed throughout India, Thailand, Vietnam, Malaysia, Indonesia, and southern China (*Peigler & Wang, 1996*). Some Taiwanese saturniines exhibited strong genetic relationships with saturniine lineages in southern China, including *Rho. verecunda*, *Act. nin. ningtaiwana*, *Sat. pyretorum*, and *Sam. wangi* (Fig. 7). Moreover, periodical Pleistocene glaciations may have led to the repeated colonization of *Act. nin. ningtaiwana* in Taiwan, as indicated by the coexistence of two divergent lineages in Taiwan (Figs. 4 and 7). However, artificial genetic admixtures occurring in *Sat. pyretorum* may explain the phenomenon of mixing genetic lineages. Because of a history of domestication for silk production, a genetic admixture between *Sam. ricini* and *Sam. canningi* (Hutton) as well as between *Ant. pernyi* and *Ant. roylei* (*Peigler & Naumann, 2003*) (Figs. S11 and S13) was evident; additionally, the domesticated hybrid *Ant. proylei* was reported to likely be derived from *Ant. roylei* and *Ant. pernyi* (*Nagaraju & Jolly, 1986*).

To address the speciation events of Lepidopteran *Heliconius erato* in the Amazonian region, *Brower (1994)* applied COI divergences and estimated the separation time to be 1.5–2 Mya. In the model insect of *Drosophila*, *Coyne & Orr (1997)* indicated that the time required for full reproductive isolation was generally 1.5–3.5 Mya. The application of COI sequences in molecular calibration has demonstrated that average speciation times were 1–2 Mya for the temperate biota of Europe and North America and approximately 2.6 Mya for the tropical Amazonian biota (*Avise, 2000*; *Hewitt, 2000*; *Knowles, 2000*; *Rull, 2008*; *Hoorn et al., 2010*; *Rull, 2011*). Studies including COI sequences have reported that the split time for most insects in Taiwanese speciation events was 1.0–2.8 Mya in the Early Pleistocene (*Yeh et al., 2004*; *Huang & Lin, 2010*; *Tsai, Wan & Yeh, 2014*; *Tsai & Yeh, 2016*; *Weng, Yang & Yeh, 2016*), although a few cases can be dated back to approximately 4 Mya (*Weng, Yang & Yeh, 2016*) or <0.3 Mya (*Weng, Yeh & Yang, 2016*) in the Middle Pleistocene.

The present study revealed that the split time for most endemic Taiwanese saturniine species and subspecies was 0.5–1.7 Mya in the Pleistocene. The inconsistency between the morphological characteristics and molecular divergence of the insular *Sat. jap. arisana* and *Rho. verecunda* may be attributable to genetic drift or local adaption when geographic discontinuity occurred in the absence of a land bridge (*Yeh et al., 2004*; *Tsao & Yeh, 2008*; *Wu et al., 2010*; *Tsai, Wan & Yeh, 2014*; *Tsai & Yeh, 2016*; *Weng, Yang & Yeh, 2016*). Moreover, this study was based on small datasets, which might possibly present only a partial differentiation history of Taiwanese saturniines. The differentiation history

of each Taiwanese satruniid will be clearer, if genomic information is included in future analyses. However, studies over the past three decades have included small datasets and have contributed critical findings in systematics.

## CONCLUSIONS

The COI nucleotide divergences within species were generally <2%, except for a few comparisons between allopatric populations that were >2%. Refugium formation during Pleistocene glaciations may have been the major driving force of biota speciation in the temperate regions of the Northern Hemisphere, and pre-Quaternary origin may be implicated for tropical Amazonian taxa. In the present study, all 16 saturniines on the subtropical continental island of Taiwan, with congeners of populations, subspecies, or species in Southeast and East Asia, exhibited a unique differentiation patterns that were less likely to have occurred than the patterns for the aforementioned temperate and tropical regions. Molecular clock dating revealed that Taiwanese endemic saturniines splitted from closely related or sibling Asian lineages primarily 0.1–1.7 Mya; each saturniine was shown to have deep, middle, or shallow genetic divergences, possibly occurring in the Early, Middle, or Late Pleistocene. The results also indicated that some montane insular saturniines might have originated from the temperate regions in Northeast Asia, whereas the other hill-distributed ones might have originated from southern China and Southeast Asia.

## ACKNOWLEDGEMENTS

We thank Yi-Sheng Tsai, Yi-Shen Chen, Wei-Chih Tsao, Yung-Jen Lu, Tzung-Hung Yang, and I-Tse Lue for collecting materials and Li-Cheh Shih for providing valuable references. We extend our gratitude to Stefan Naumann and Hazel Davies for editing and discussing this manuscript. We would also like to acknowledge the technical services provided by the Sequencing Core Facility of the National Yang-Ming University Genome Research Center, which is supported by the National Science Council.

### Funding

This work was supported by the National Science Council of Taiwan (NSC 97-2317-B-005-008 and 98- 2321-B-005-011) and partially supported by the Vietnam National Foundation for Science and Technology Development (NAFOSTED) under the grant number 106-NN.05-2016.04, Vietnam. The funders had no role in study design, data collection and analysis, decision to publish, or preparation of the manuscript.

### Grant Disclosures

The following grant information was disclosed by the authors:
National Science Council of Taiwan: NSC 97-2317-B-005-008 and 98-2321-B-005-011.
Vietnam National Foundation for Science and Technology Development (NAFOSTED): 106-NN.05-2016.04.

## Competing Interests

The authors declare that they have no competing interests.

## Author Contributions

- Wen-Bin Yeh conceived and designed the experiments, analyzed the data, prepared figures and/or tables, authored or reviewed drafts of the paper, and approved the final draft.
- Cheng-Lung Tsai conceived and designed the experiments, analyzed the data, prepared figures and/or tables, authored or reviewed drafts of the paper, and approved the final draft.
- Thai-Hong Pham performed the experiments, authored or reviewed drafts of the paper, and approved the final draft.
- Shipher Wu performed the experiments, authored or reviewed drafts of the paper, and approved the final draft.
- Chia-Wei Chang performed the experiments, analyzed the data, authored or reviewed drafts of the paper, and approved the final draft.
- Hong-Minh Bui conceived and designed the experiments, authored or reviewed drafts of the paper, and approved the final draft.

## Field Study Permissions

The following information was supplied relating to field study approvals (*i.e.*, approving body and any reference numbers):

Field experiments were approved by Taroko National Park (permission number 201202200200, 201303080267, 201402270306); Yushan National Park (permission number 1040000498); and Shei-Pa National Park (permission number 1041000097).

## DNA Deposition

The following information was supplied regarding the deposition of DNA sequences:

The saturniid sequences are available at GenBank: AB533547 to AB533597 for the COI gene, AB841436 to AB841480 for 16S rRNA gene, AB841488 to AB841532 for 28S rRNA gene.

## Data Availability

The raw data are available in the Supplemental Files.

## Supplemental Information

Supplemental information for this article can be found online at http://dx.doi.org/10.7717/peerj.13240#supplemental-information.

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
