# Peer review of "Differentiation patterns of emperor moths (Lepidoptera: Saturniidae: Saturniinae) of a continental island: divergent evolutionary history driven by Pleistocene glaciations"

_PeerJ, doi:10.7717/peerj.13240_

## Round 0.1 · original submission · Major Revisions

I have received two reports for your study and both suggest revisions. Please pay close attention to their comments in your revision, especially to the following:

1. phylogenetic inference method highlighted,
2. molecular timing rate estimations,
3. language and syntax,
4. considering your results in the context of what is already known for the discussion.

Reviewer 1 ·

Basic reporting

The authors of "Differentiation patterns of emperor moths (Lepidoptera: Saturniidae) of a continental island: divergent evolutionary history driven by Pleistocene glaciations" present an interesting paper on the genetic distances between moths in Taiwan and their mainland sisters, with dated phylogenies to place the timing of splits in the Pleistocene. The molecular data consisted of three genes (of which one was mostly invariable), which is considered a small dataset in todays world of genomic sequencing options, but the findings are worth publication regardless, at the least as a basis for further research.

The manuscript at present needs a fair bit of 'polishing'. Quite strikingly, the english language needs to be checked for grammar and spelling - there are sentences that are incomprehensible. That said, I was able to perform the review to find the following issues that should be addressed.

It is worth pointing out in the introduction that Saturniidae, despite their large size, are poor flyers and will have to rely on land bridges to disperse, as opposed to some other large moths - e.g. sphingids. This is an important underpinning for the study.

Lines 187-8: Neighbor-joining is a 'quick and dirty' method for phylogenetic inference, not suited for phylogenetic publications because repeats can give different outcomes. Use a maximum likelihood approach instead.

For data retrieved from GenBank: please include how you ensured data integrity, how can you be sure of the accuracy of the identifications?

This study by Merckx et al. 2015 covers a quite similar topic and is perhaps worth comparing the author's data against:

Merckx VSFT, Hendriks KP, Beentjes KK, Mennes CB, Becking LE, Peijnenburg KTCA, Afendy A, Arumugam N, de Boer H, Biun A, Buang MM, Chen P-P, Chung AYC, Dow R, Feijen FAA, Feijen H, Soest CF, Geml J, Geurts R, Gravendeel B, Hovenkamp P, Imbun P, Ipor I, Janssens SB, Jocqué M, Kappes H, Khoo E, Koomen P, Lens F, Majapun RJ, Morgado LN, Neupane S, Nieser N, Pereira JT, Rahman H, Sabran S, Sawang A, Schwallier RM, Shim P-S, Smit H, Sol N, Spait M, Stech M, Stokvis F, Sugau JB, Suleiman M, Sumail S, Thomas DC, van Tol J, Tuh FYY, Yahya BE, Nais J, Repin R, Lakim M, Schilthuizen M (2015) Evolution of endemism on a young tropical mountain. Nature 524: 347–350. https://doi.org/10.1038/nature14949


Given the source and size of the data used for the study, the discussion should incorporate a section with recent papers on the dangers of using small (mitochondrial) datasets and molecular dating. See e.g:

Ebdon S, Laetsch DR, Dapporto L, Hayward A, Ritchie MG, Dinca V, Vila R & Lohse K (2021) The Pleistocene species pump past its prime: evidence from European butterfly sister species. Molecular Ecology (published online ahead of print). DOI: 10.1111/mec.15981

I support that the authors did not use their data for any formal nomenclatural changes.

With the aforementioned issues addressed, I recommend this manuscript for publication.

Experimental design

no comment

Validity of the findings

no comment

Additional comments

no comment

Reviewer 2 ·

Basic reporting

The article is in general clearly written, although there are some grammatical mistakes that could be cleaned up. I did not keep track of them.

Experimental design

This is a very interesting topic, and certainly worthy of investigation. The authors have looked at the divergences of species of Saturniidae moths in Taiwan compared to conspecifics or congeners on the mainland. They base their findings on three loci (2 mitochondrial and one nuclear). I have to say I was a bit surprised by their choice of markers, COI is obvious, but why 16S and 28S? If one looks at the Lepidoptera literature, there are a number of other widely used markers (EF1a, wingless, GAPDH, RpS5, etc) that would have made more sense to sequence in a PCR based study. What's done is done however.

Another surprise was that the authors chose to do a NJ analysis for the combined dataset, as well as a Bayesian analysis. If I understood correctly, the tree shown in Fig 2 is the NJ tree, why is that? It has been shown time and again that NJ is not a particularly good method for estimating phylogenetic relationships, and it would be preferable to report the results of a maximum likelihood and/or Bayesian analysis. The deeper nodes seem to have no support from the Bayesian analysis, is that because the topology was different? Outgroups are reported to be saturniids in related subfamilies, however the tree in Fig 2 has Bombyx mori and the sphingid Theretra, distant relatives to Saturniidae (with 3 markers, maybe too distant). I would suggest rerunning the phylogenetic analyses with IQ-Tree and MrBayes with appropriate outgroups.

Finally, the use of a molecular clock rate for the timing of divergence analyses, especially a rate that was estimated 25 years ago based on minimal data, is perhaps not the way to do things in the 2020s. Why have the authors not attempted to estimate times of divergence based on calibrations from published studies? E.g. Kawahara et al (2019, PNAS) include Attacus, Saturnia, Actias, and Hemileucinae representatives. Their estimates for times of divergence of these lineages can be used as secondary calibrations in the authors own dataset to get estimates of COI rates for each genus separately. Very easy to get these values from BEAST. This would not change the qualitative results of the different lineages having different times of divergence, but the quantitative results will likely change (the times of divergence).

Validity of the findings

The main findings, that different lineages have different histories, with some lineages diverging from mainland lineages a long time ago, vs those lineages that are still very close to mainland lineages, will probably remain after re-analysis of the data. This is an interesting result, expected to some degree, but good to show. There are also some surprises with e.g. the Attacus lineages being so divergent form each other.

---

## Round 0.2 · Minor Revisions

You have addressed many of the comments raised by the reviewers, but there are several concerns raised by one of the reviewers. Please address these.

Reviewer 1 ·

Basic reporting

This is my second review of the manuscript titled "Differentiation patterns of emperor moths (Lepidoptera: Saturniidae) of a continental island: divergent evolutionary history driven by Pleistocene glaciations".

Overall, the authors have accurately addressed the previously raised concerns.

I noticed however that the study is restricted to the subfamily Saturniinae (line 181), but througout the manuscript the authors refer to their study subjects as "saturniids" or "saturniid species", which would refer to the family level. To be more precise, the adjectival form of Saturniinae is saturniines or saturniine species.

line 63: "evolved with a unique differentiation pattern" - can you be more specific? Are we talking about timing, geneflow, geographic connections or more theoretical concepts like allopatry/parapatry etc? And how are they specifically different from those in the Amazonian regions?

The aforementioned suggestions should be easy to incorporate and with those I recommend the manuscript be accepted for publication.

Experimental design

no comment

Validity of the findings

no comment

Additional comments

no comment

Reviewer 2 ·

Basic reporting

The revision is a much improved version of the manuscript. The grammar is now fine. Spelling of scientific names should be double checked, I noted the following misspellings:

Line 250 Antheraea superb --> superba
Lines 323+420 boisdubalii --> boisduvalii (also in Fig 7)
Line 396 aturniid --> saturniid

There may be more.

Experimental design

I accept the authors argumentation that the relaxed molecular clock using secondary calibrations did not work as expected. An interesting observation, but I guess it is known that one cannot use the same clock models for intraspecific vs interspecific analyses in the same dataset. As it is, the strict molecular clock analyses are probably in the right ball park with regard to times of divergence.

Validity of the findings

The findings are interesting and increase our understanding of how fauna develop over evolutionary time. Even if the estimated times of divergence are not exact, they still tell the story of different histories for the different taxon pairs.

---

## Round 0.3 · accepted · Accept

You have addressed the reviewers' concerns to an appreciable degree, and thus, I am happy to accept your paper. I wish you the best with your future research.